# Binarized Diffusion Model for Image Super-Resolution

**Zheng Chen[1], Haotong Qin[2]\*, Yong Guo[3], Xiongfei Su[4],**
**Xin Yuan[4], Linghe Kong[1], Yulun Zhang[1]\***
[1]Shanghai Jiao Tong University, [2]ETH Zürich,
[3]Max Planck Institute for Informatics, [4]Westlake University

## Abstract

Advanced diffusion models (DMs) perform impressively in image super-resolution (SR), but the high memory and computational costs hinder their deployment. Binarization, an ultra-compression algorithm, offers the potential for effectively accelerating DMs. Nonetheless, due to the model structure and the multi-step iterative attribute of DMs, existing binarization methods result in significant performance degradation. In this paper, we introduce a novel binarized diffusion model, BI-DiffSR, for image SR. First, for the model structure, we design a UNet architecture optimized for binarization. We propose the consistent-pixel-downsample (CP-Down) and consistent-pixel-upsample (CP-Up) to maintain dimension consistent and facilitate the full-precision information transfer. Meanwhile, we design the channel-shuffle-fusion (CS-Fusion) to enhance feature fusion in skip connection. Second, for the activation difference across timestep, we design the timestep-aware redistribution (TaR) and activation function (TaA). The TaR and TaA dynamically adjust the distribution of activations based on different timesteps, improving the flexibility and representation alability of the binarized module. Comprehensive experiments demonstrate that our BI-DiffSR outperforms existing binarization methods. Code is released at: `https://github.com/zhengchen1999/BI-DiffSR`.

## 1 Introduction

Image super-resolution (SR) is a fundamental task in low-level vision and image processing. It aims to reconstruct high-resolution (HR) images from low-resolution (LR) counterparts. Currently, the mainstream methods for this task are deep neural networks, which employ learning-based techniques to map LR images to HR images [10, 70, 31, 54, 6, 68]. Among these methods, generative models [62, 9, 44] have garnered widespread attention for their ability to restore more realism results.

Especially, the diffusion model (DM) [16, 58, 52], a newly proposed generative model, exhibits impressive performance. With its superior generation quality and more stable training, diffusion model is widely used in various image tasks, including image SR [54, 63]. Specifically, the diffusion model transforms a standard Gaussian distribution into a high-quality image through a stochastic iterative denoising process. In image SR, it further constrains the generation scope by conditioning on the LR image to produce the targeted HR image.

However, to produce high-quality results, diffusion models require thousands of iterative steps, leading to slow inference processes and high computational costs. Some methods [58, 40, 37] implement faster sampling strategies via learning sample trajectories, effectively reducing the number of iterations to tens. Yet, a single inference step still demands substantial memory usage and floating-point computations, especially for SR tasks involving high-resolution images. Meanwhile, most edge devices (*e.g.*, mobile and IoT devices), have limited storage and computational resources. This hampers the deployment of diffusion models on these platforms and limits their application. Therefore, it is essential to compress diffusion models to accelerate inference speed and reduce computational costs while maintaining model performance.

---

\*Corresponding authors: Haotong Qin, qinhaotong@gmail.com; Yulun Zhang, yulun100@gmail.com

38th Conference on Neural Information Processing Systems (NeurIPS 2024).

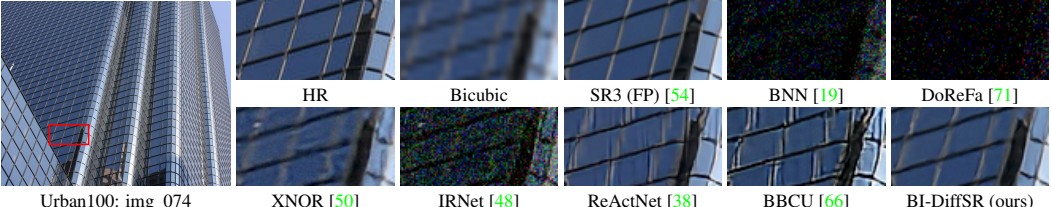

Figure 1: Visual comparison (×4) of binarization methods. Some methods (*e.g.*, BNN [19]) cannot work on diffusion models. Several methods (*e.g.*, BBCU [66]) suffer from blurring and artifacts. In contrast, our proposed BI-DiffSR outperforms other methods with accurate results.

Common compression approaches include pruning [11], distillation [61], and quantization [45, 66, 26]. Among these, 1-bit quantization (*i.e.*, binarization) stands out for its effectiveness. As the most aggressive form of bit-width reduction, binarization significantly reduces memory and computational costs by quantizing the weights and activations of full-precision (32-bit) models to 1-bit.

Nonetheless, existing binarization research primarily deals with higher-level tasks (*e.g.*, classification) and end-to-end models [49, 19, 39]. Applying existing binarization methods directly to current diffusion model architectures results in a significant performance drop. This is primarily due to two aspects: **(1) Model Structure.** Diffusion models typically apply the UNet architecture [53] for noise estimation, which is not easy to binarize directly. *I. Dimension Mismatch:* The identity shortcut is crucial for the binarized SR model, since it facilitates the transfer of full-precision (FP) information, compensating for the binarized model [66]. However, in UNet, the feature dimensions change since downsampling/upsampling. The dimension mismatch prevents the usage of shortcuts, cutting off the full-precision propagation. *II. Fusion Difficulty:* The UNet structure uses skip connections to transfer information from encoder to decoder. However, the typical fusion method, concatenation, leads to the dimension mismatch. Alternatively, other methods (*e.g.*, addition) also struggle to achieve effective fusion due to significant differences in value ranges between encoder and decoder features. **(2) Activation Distribution.** Due to the multi-step iterative nature of diffusion models, the activation distribution dramatically changes with timesteps. Furthermore, the activation binarization will even amplify activation differences [50]. The difference increases the learning challenges for binarized modules (*e.g.*, binarized convolution), thereby hindering the effective representation of features. Consequently, the SR performance of the binarized diffusion model is limited.

Based on the above analysis, we propose a novel binarized diffusion model, BI-DiffSR, to achieve effective image SR. Our design comprises two main aspects: structure and activation. **(1) Structure.** We develop a simple yet effective convolutional UNet architecture, which is suitable for binarization. *I. Dimension Consistency:* We propose consistent-pixel-downsample (CP-Down) and consistent-pixel-upsample (CP-Up) to ensure dimensional consistency in binarized computation. CP-Down and CP-Up maintain the full-precision information transfer during feature scaling. *II. Feature Fusion:* We develop the channel-shuffle-fusion (CS-Fusion) to facilitate the fusion of different features within skip connections and suit binarized modules. Through channel shuffle, we combine two input features into two shuffled features to balance their activation value ranges. **(2) Activation.** Considering the activation differences at different timesteps, we design the timestep-aware redistribution (TaR) and timestep-aware activation function (TaA). The TaR and TaA adjust the binarized module input and output activations according to different timesteps. This timestep-aware adjustment improves the flexibility and representational ability of the binarized module to various activation distributions.

Extensive experiments demonstrate that our proposed BI-DiffSR significantly outperforms existing binarization methods. As shown in Fig. 1, our BI-DiffSR restores more perceptually pleasing results than other methods. Overall, our contributions are as follows:

- We design the novel binarized model, BI-DiffSR, for image SR. To the best of our knowledge, this is the first binarized diffusion model applied to SR.

- We develop a UNet architecture optimized for binarization, with consistent-pixel-downsample (CP-Down) and upsample (CP-Up), and channel-shuffle-fusion (CS-Fusion).

- We introduce the timestep-aware redistribution (TaR) and activation function (TaA) to adapt activation distributions by timestep, enhancing the capabilities of the binarized module.

- Our BI-DiffSR surpasses current state-of-the-art binarization methods, and offers comparable perceptual performance to full-precision diffusion models.

## 2  Related Work

### 2.1  Image Super-Resolution

Since the advent of SRCNN [10], deep neural networks have gradually become the mainstream for image SR. Numerous architectures [33, 70, 46, 31, 5] are designed to advance reconstruction accuracy. Concurrently, generative methods are widely applied to improve the quality of restored image details. This includes autoregressive model [23, 9], normalizing flow [51, 41, 32], and generative adversarial network (GAN) [13, 24]. For instance, SRFlow [41] utilizes normalizing flows to transform a Gaussian distribution into the HR image space. Meanwhile, SRGAN [24] employs GAN as supervision loss and combines it with perceptual loss to produce visually pleasing results. Subsequent methods [62, 4] further refine the network and loss to yield more natural results. Recently, the diffusion model (DM) [16, 8] has been introduced into SR, displaying impressive performance, especially regarding perception. Thereby, DM has been attracting widespread attention [54, 25, 65].

### 2.2  Diffusion Model

Through the Markov chain, the diffusion model (DM) generates images from the Gaussian distribution [16]. It has demonstrated exceptional performance in various tasks [3, 17, 52, 7, 14, 30, 29, 36, 35, 28, 15]. Naturally, DM has also been extensively researched in the field of image SR [54, 21, 63, 34, 65]. For instance, SR3 [54] achieves conditional diffusion by concatenating the resized LR image with the noise image as the input of the noise estimation network. Meanwhile, some methods, *e.g.*, DDNM [63], utilize an unconditional pre-trained diffusion model as a prior for zero-shot SR. Additionally, some approaches [34, 65] employ text-to-image diffusion models to achieve realistic and controllable SR. Despite promising results, these methods require hundreds or thousands of sampling steps to generate HR images. Although some acceleration algorithms [58, 37, 28] reduce the inference steps to tens, each denoising step still demands substantial resources. The high memory and computational costs hinder the practical application of DMs on resource-limited platforms (*e.g.*, mobile devices). To address this issue, we design a practical binarized SR diffusion model.

### 2.3  Binarization

Binarization is a popular model compression approach [49]. As an extreme case of quantization, it reduces the weights and activations of a full-precision neural network to 1-bit. This significantly decreases the model size and computational complexity, making it widely used in both high-level [19, 39, 48, 38, 67] and low-level [20, 66, 66, 69] vision tasks. For example, BNN [19] directly binarizes weights and activations during forward and backward processes. IRNet [48] retains information accurately through the proposed information retention network. ReActNet [38] proposes the RSign and RPReLU to enable explicit distribution reshape and shift of activations. Meanwhile, in the image SR field, BBCU [66] introduces a meticulously designed basic binary conv unit, which removes batch normalization (BN) in the binarized model. However, for DM, though some methods realize low-bit (*e.g.*, 4 or 8) quantization [55, 26, 27], implementing binarization remains challenging. Due to the structure of the noise estimation network and the multi-step iterative attribute, existing binarization methods often result in significant SR performance degradation.

## 3  Method

In this section, we introduce our proposed BI-DiffSR. First, we describe the structural designs suitable for binarization: *consistent-pixel-downsample* (CP-Down), *consistent-pixel-upsample* (CP-Up), and *channel-shuffle-fusion module* (CS-Fusion). The CP-Down and CP-Up achieve dimension adjustment and ensure the transfer of full-precision information. The CS-Fusion effectively integrates different features within the skip connection. Secondly, we present the dynamic designs tailored for varying activations: *timestep-aware redistribution* (TaR) and *activation function* (TaA). The TaR and TaA enhance the representational learning of the binarized modules across multiple timesteps.

### 3.1  Model Structure

**Overall.** We employ a convolutional UNet [53] as the noise estimation network. Details of the diffusion model for SR are provided in the supplementary materials. As the common choice within DMs, using UNet as the backbone for binarization offers generalizability. Moreover, for binarized models, the design should be compact and well-defined. Compared to the non-local self-attention operations, convolution is simpler and easier to implement. Our architecture is shown in Fig. 2a, featuring an encoder-bottleneck-decoder ($\mathcal{E}$-$\mathcal{B}$-$\mathcal{D}$) design.

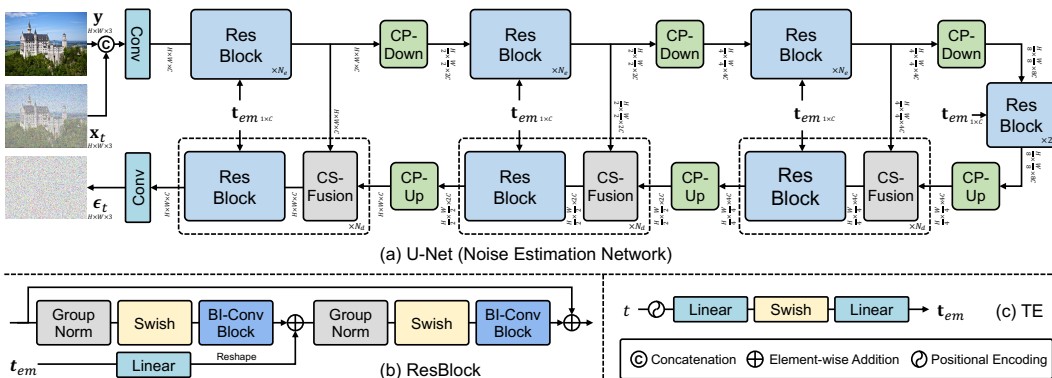

Figure 2: The overall structure of the noise estimation network. (a) UNet: The model consists of ResBlock, CP-Down, CP-Up, and CS-Fusion. It predicts noise $\boldsymbol{\epsilon}_t$ with the upscaled LR image $\mathbf{y}$, noise image $\mathbf{x}_t$, and timestep $t$. (b) ResBlock: Residual block, utilizes the binarized convolution (BI-Conv) block. The input and output dimensions of the block remain consistent, making it suitable for binarization. (c) TE: Time encoding, encoders timestep $t$ to produce the timestep embedding $\mathbf{t}_{em}$.

Given the noise image $\mathbf{x}_t \in \mathbb{R}^{H \times W \times 3}$ at $t$-th timestep, and the LR image $\mathbf{y} \in \mathbb{R}^{H \times W \times 3}$ (bicubic to HR resolution), two images are concatenated along the channel dimension as the UNet input, where $H \times W$ is the resolution. For timestep $t$, the sinusoidal position encoding [60] is applied to obtain the timestep embedding $\mathbf{t}_{em} \in \mathbb{R}^C$. The input images first pass through a convolutional layer to produce the shallow feature $\mathbf{F}_s \in \mathbb{R}^{H \times W \times C}$, where $C$ is the channel number. Then, the shallow feature $\mathbf{F}_s$ are further refined by the $\mathcal{E}$-$\mathcal{B}$-$\mathcal{D}$ into the deepe feature $\mathbf{F}_d \in \mathbb{R}^{H \times W \times C}$. Each level of the $\mathcal{E}$-$\mathcal{B}$-$\mathcal{D}$ is composed of multiple ($N_e$ in $\mathcal{E}$ and $N_d$ in $\mathcal{D}$) residual blocks (ResBlocks), with details illustrated in Fig. 2b. Within the ResBlocks, the timestep embedding $\mathbf{t}_{em}$ is incorporated to provide temporal information. In the encoder $\mathcal{E}$, downsample module (*i.e.*, CP-Down) progressively reduces feature resolution and increases channel number. Conversely, in the decoder $\mathcal{D}$, upsample module (*i.e.*, CP-Up) gradually restores the high-resolution representation. Moreover, to compensate for information loss during downsampling, the skip connection is used to link features between the encoder and decoder. Finally, through one convolution, the predicted noise $\boldsymbol{\epsilon}_t \in \mathbb{R}^{H \times W \times 3}$ is obtained.

**Structure Analysis.** Although the UNet architecture is suitable for diffusion models, its unique structure poses challenges for direct binarization, which results in a substantial accuracy decrease compared to full-precision models. We identify two main issues/challenges that contribute to the problem: ***dimension mismatch*** and ***fusion difficulty***.

***Challenge I: Dimension Mismatch.*** In the binarized model, 1-bit quantization leads to significant information loss, limiting the capability for feature representation and the ultimate SR performance. Compared to binary activations, full-precision activations contain more information. Therefore, we can apply the identity shortcut to preserve the full-precision information. This operation effectively compensates for the information loss caused by binarization. However, in UNet, the frequent changes in feature resolution and channel size lead to dimension mismatches. This prevents the effective use of the identity shortcut and cuts off the propagation of full-precision information.

***Challenge II: Fusion Difficulty.*** Another crucial structure of UNet is the skip connection, which links encoder and decoder features. The typical approach is to concatenate these features along the channel dimension and pass them to subsequent layers. However, concatenate causes dimension mismatch. As analyzed in ***Challenge I***, it is unsuitable for binarization. Furthermore, we find that there is a significant difference in the activation ranges between the two inputs (from encoder and decoder) of the skip connection (Fig. 3d). This imbalance makes other fusion methods, *e.g.*, addition, also unsuitable, since the smaller range activation is masked by the larger one, as illustrated in Fig. 3d.

To better adapt binarization for the UNet architecture, we propose two structures: ***Consistent-Downsample/Upsample*** and ***Channel-Shuffle Fusion***, as illustrated in Fig. 3.

**Consistent-Pixel-Downsample/Upsample.** To address the dimension mismatch in the UNet structure, we first confine all feature reshaping operations to the Upsample and Downsample modules. That is to ensure that the dimension of the main module, *i.e.*, ResBlock, remains matched. Meanwhile, we propose the consistent-pixel-downsample (CP-Down) and consistent-pixel-upsample (CP-Up).

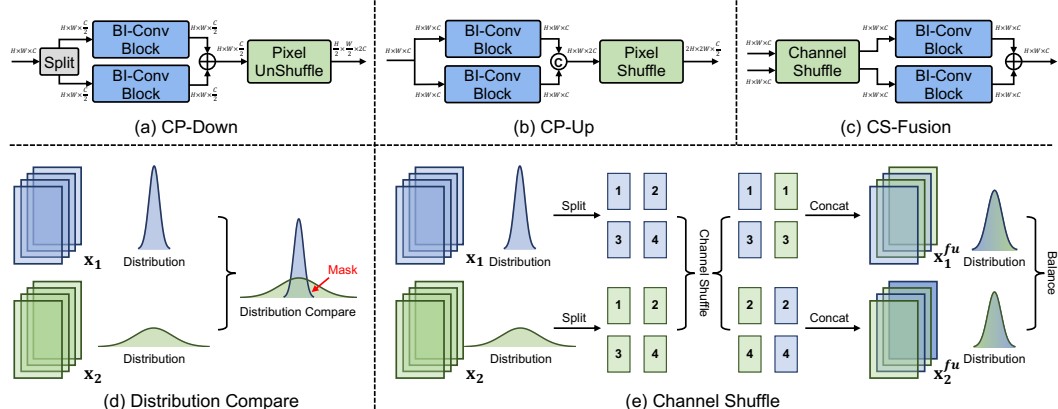

Figure 3: (a) CP-Down: Consistent-pixel-downsample. (b) CP-Up: Consistent-pixel-upsample. (c) CS-Fusion: Channel-shuffle fusion. (d) In the skip connection, the value ranges of two features ($\mathbf{x}_1$, $\mathbf{x}_2$) may be significant differences, which impedes effective fusion. (e) The illustration of channel shuffle. the shuffled features ($\mathbf{x}_1^{sh}$, $\mathbf{x}_2^{sh}$) have closely matched value ranges.

*(1) CP-Down:* We evenly split the input features $\mathbf{x}_{in}^{do} \in \mathbb{R}^{H \times W \times C}$ along the channel dimension and process them through two convolutions with identical input and output dimensions. The stable (matching) dimension allows the usage of identity shortcuts. Finally, by applying Pixel-UnShuffle [57], we reduce the resolution of the features while increasing the channel number. The formula is:

$$\mathbf{x}_{in}^{do} = [\mathbf{x}_s^1, \mathbf{x}_s^2], \quad \mathbf{x}_s^i \in \mathbb{R}^{H \times W \times \frac{C}{2}}, \quad \mathbf{x}_{out}^{do} = \mathcal{PS}^{-1}\left(\mathcal{C}_1(\mathbf{x}_s^1) + \mathcal{C}_2(\mathbf{x}_s^2)\right), \quad (1)$$

where $\mathbf{x}_{out}^{do} \in \mathbb{R}^{\frac{H}{2} \times \frac{W}{2} \times 2C}$ is the output of CP-Down; $\mathcal{C}_1(\cdot)$ and $\mathcal{C}_2(\cdot)$ represent two (binarized) convolutions; $\mathcal{PS}^{-1}$ denotes the Pixel-UnShuffle operation.

*(2) CP-Up:* Similarly, feature upsampling is achieved through two convolutions combined with Pixel-Shuffle. The operation can be mathematically expressed as follows:

$$\mathbf{x}_{out}^{up} = \mathcal{PS}\left(\text{Concat}\left(\mathcal{C}_1\left(\mathbf{x}_{in}^{up}\right), \mathcal{C}_2\left(\mathbf{x}_{in}^{up}\right)\right)\right), \quad (2)$$

where, $\mathbf{x}_{in}^{up} \in \mathbb{R}^{H \times W \times C}$ and $\mathbf{x}_{out}^{up} \in \mathbb{R}^{2H \times 2W \times \frac{C}{2}}$ denotes the input and output of CP-Up; $\text{Concat}(\cdot)$ represents the channel concatenation operation; $\mathcal{PS}$ is the Pixel-Shuffle operation.

With the above design, we ensure the flow of full-precision information throughout the UNet, effectively improving feature representation and enhancing SR performance.

**Channel-Shuffle Fusion.** To effectively fuse the features in the skip connection while meeting the requirements for dimension matching in binarization, we propose the channel-shuffle fusion (CS-Fusion), as shown in Fig. 3c. Given two features $\mathbf{x}_1$, $\mathbf{x}_2 \in \mathbb{R}^{H \times W \times C}$, we first employ the channel-shuffle operation to mitigate the differences in their value ranges, as illustrated in Fig. 3e. Specifically, we split the two features according to the odd and even channel indexes. Then, we pair and concatenate features along the channel dimension, based on odd and even indexes, to produce two new shuffle features $\mathbf{x}_1^{sh}$, $\mathbf{x}_2^{sh} \in \mathbb{R}^{H \times W \times C}$. This process can be formulated as follows:

$$\mathbf{x}_n = [\mathbf{x}_n^1, \mathbf{x}_n^2, \ldots, \mathbf{x}_n^{C-1}, \mathbf{x}_n^C], \quad n \in \{1, 2\},$$
$$\mathbf{x}_m^{sh} = \text{Concat}\left(\left\{\mathbf{x}_j^{2i+(m-1)} \mid i = 1, \ldots, C/2, j = 1, 2\right\}\right), \quad m \in \{1, 2\}, \quad (3)$$

Through visualization in Fig. 3e, we can observe that the value range of features after channel shuffle becomes balanced. Subsequently, we process the shuffled features through two convolutions and addition to produce the final fused feature $\mathbf{x}_{out}^{sh} \in \mathbb{R}^{H \times W \times C}$, in a manner similar to Eq. (1), as:

$$\mathbf{x}_{out}^{sh} = \mathcal{C}_1^{sh}(\mathbf{x}_1^{sh}) + \mathcal{C}_2^{sh}(\mathbf{x}_2^{sh}), \quad (4)$$

where $\mathcal{C}_1^{sh}(\cdot)$ and $\mathcal{C}_2^{sh}(\cdot)$ are two (binarized) convolutions. This process realizes the fusion of two features, ensuring that dimensions are matched within the fusion process and in subsequent modules (*e.g.*, ResBlock). Meanwhile, the matched dimension allows the usage of the identity shortcut, thus effectively transferring full-precision information. Overall, our proposed CS-Fusion achieves effective feature integration in the skip connection. Therefore, the binarized model can better represent features and improve SR performance. Furthermore, our CS-Fusion does not introduce additional memory or computational overhead since the channel shuffle only involves feature transformation operations. Experiments in Sec. 4.2 further reveal the impacts of CS-Fusion.

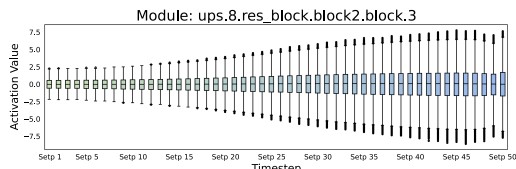 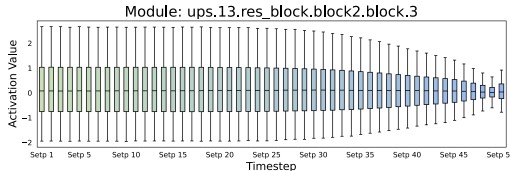

Figure 4: Visualization of the changes in activation distribution across 50 timesteps.

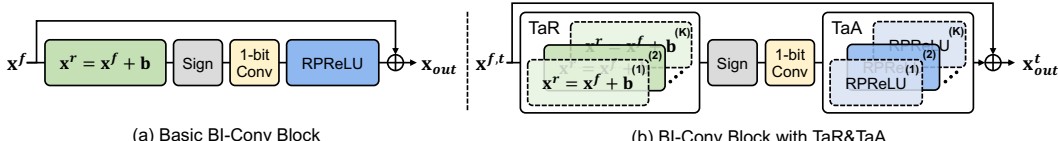

(a) Basic BI-Conv Block                    (b) BI-Conv Block with TaR&TaA

Figure 5: (a) The basic binarized convolutional (BI-Conv) block. The learnable bias $\mathbf{b}$ and the activation function RPReLU adjust the activations. (b) In timestep-aware redistribution (TaR) and activation function (TaA), multiple pairs of $\mathbf{b}$ and RPReLU are applied to adapt to the multi-step in DM. At each step $t$, only one pair of $\mathbf{b}$ and RPReLU is used (the darker modules with solid lines).

## 3.2  Activation Distribution

**Basic Binarized Convolutional Block.** We first introduce the basic binarized module, as illustrated in Fig. 5a. For the full-precision activation $\mathbf{x}^f \in \mathbb{R}^{H \times W \times C}$, we initially shift its distribution and binarize the shifted activation to 1-bit activations with sign function $\text{Sign}(\cdot)$. The process is:

$$\mathbf{x}^r = \mathbf{x}^f + \mathbf{b}, \quad x^b = \text{Sign}\left(x^r\right) = \begin{cases} +1, & x^r \geq 0 \\ -1, & x^r < 0 \end{cases} \left(\forall x^r \in \mathbf{x}^r,\ \forall x^b \in \mathbf{x}^b\right), \quad (5)$$

where $\mathbf{b} \in \mathbb{R}^C$ is a learnable parameter; $\mathbf{x}^b \in \mathbb{R}^{H \times W \times C}$ is the 1-bit activation. Meanwhile, for the binarized convolution, the full-precision weight $\mathbf{w}^f \in \mathbb{R}^{C_{out} \times C_{in} \times K_h \times K_w}$ is also binarized to 1-bit weight $\mathbf{w}^b \in \mathbb{R}^{C_{out} \times C_{in} \times K_h \times K_w}$. To compensate for the differences between binary and full-precision weights, we scale $\mathbf{w}^b$ using the mean absolute value of $\mathbf{w}^f$ [50]. The total operation is:

$$w^b = \frac{\left\|\mathbf{w}^f\right\|_1}{n} \cdot \text{Sign}(w^f), \quad \forall w^f \in \mathbf{w}^f,\ \forall w^b \in \mathbf{w}^b, \quad (6)$$

where $n$ is the number of $\mathbf{w}^f$ values. Subsequently, the floating-point matrix multiplication in full-precision convolution can be replaced by logical XNOR and bit-counting operations as:

$$\mathbf{x}_{out}^b = \mathbf{x}^b * \mathbf{w}^b = \text{bit-count}\left(\text{XNOR}\left(\mathbf{x}^b, \mathbf{w}^b\right)\right) \quad (7)$$

where $*$ means the convolutional operation; $\mathbf{x}_{out}^b \in \mathbb{R}^{H \times W \times C}$ is the output of 1-bit convolution. Then, we adjust $\mathbf{x}_{out}^b$ with the activation function RPReLU [38], resulting in $\mathbf{x}_{act}^b \in \mathbb{R}^{H \times W \times C}$.

Finally, we combine $\mathbf{x}_{act}^b$ with full-precision activation $\mathbf{x}^f$ via an identity shortcut to get the final output $\mathbf{x}_{out} \in \mathbb{R}^{H \times W \times C}$. Moreover, since the sign function $\text{Sign}(\cdot)$ is non-differentiable, we use the straight-through estimator (STE) [1] for backpropagation to train binarized models.

**Distribution Analysis.** In diffusion models, the multi-step iterative design leads to changes in the activation distribution as the timestep changes. By visualizing the activation distributions at different timesteps in Fig. 4, we can observe that activation distributions of adjacent timesteps are similar, whereas those separated by larger intervals show significant differences.

For full-precision models, the impact of these variations may be small due to the real-valued weight and activation. In contrast, for binarized modules, the activation distribution has a substantial impact on feature representation, and consequently, affects the SR performance. This is because 1-bit modules, due to the binary weights, struggle to effectively learn representations from different distributions, thereby limiting their modeling capabilities. Meanwhile, during the activation binarization, the sign function further amplifies activation differences, particularly for values around zero [38].

The basic binarized module utilizes the learnable biase and the activation function RPReLU to adjust the input and output activations. This approach mitigates the representational challenges posed by activation distribution differences across timestep to some extent. However, these static designs are insufficient to cope with the extreme activation changes across multiple timesteps in diffusion models. Consequently, the SR performance of the binarized diffusion model is limited. Experiments in Sec. 4.2, further demonstrate the above analyses.

**Timestep-aware Redistribution/Activation Function.** To cope with the variability of activation distribution with timestep, we propose the timestep-aware redistribution (TaR) and timestep-aware activation function (TaA). The module details are illustrated in Fig. 5b. The design of TaR and TaA is inspired by the mixture of experts (MoE) [56], applying a set of learnable biases and RPReLU activation functions to accommodate different timesteps.

Specifically, we apply $K$ pairs of bias and RPReLU for TaR ($\mathbf{b}^{(i)}\in\mathbb{R}^C$) and TaA (RPReLU$^{(i)}$), where $i\in\{1, 2, \ldots, K\}$. Given the total timesteps (*e.g.*, $\{1, 2, \ldots, T\}$), we evenly divide them into $K$ groups in sequence. For the input activation $\mathbf{x}^{f,t}\in\mathbb{R}^{H\times W\times C}$ at $t$-th timstep ($t\in\{1, 2, \ldots, T\}$), we select the corresponding pair of bias and RPReLU based on the group associated with $t$, to adjust its input and output activation. The process can be formulated as:

$$
\mathbf{x}^{r,t} = \mathrm{TaR}(\mathbf{x}_{in}^t) = \mathbf{x}_{in}^t + \sum_{i=1}^{K} \mathbf{1}_{i=\lfloor K\times t/T\rfloor} \cdot \mathbf{b}^{(i)},
$$
$$
\mathbf{x}_{act}^{b,t} = \mathrm{TaA}(\mathbf{x}_{out}^{b,t}) = \sum_{i=1}^{K} \mathbf{1}_{i=\lfloor K\times t/T\rfloor} \mathrm{RPReLU}^{(i)}(\mathbf{x}_{out}^{b,t}),
$$

(8)

where $\mathbf{1}_{(\cdot)}$ is the indicator function; $\mathbf{x}^{r,t}$, $\mathbf{x}_{out}^{b,t}$, $\mathbf{x}_{act}^{b,t}\in\mathbb{R}^{H\times W\times C}$, represent, at $t$-th timestep, the shifted input activation, the output of the 1-bit convolution, the output of the RPReLU activation function, respectively. Since the activations at adjacent timesteps exhibit a certain degree of similarity (as shown in Fig. 4), we employ the fixed grouping sampling strategy (defined in Eq. (8)).

Essentially, the TaR and TaA segment the multi-step process into smaller groups, limiting the range of activation changes. This reduces the difficulty of adjusting activations, allowing the binarized module to better adapt to changing activations. Therefore, the proposed TaR and TaA can effectively enhance the representation ability of the binarized module and ultimately improve SR performance. Meanwhile, compared to the basic module, there are no additional computational costs in our TaR and TaA. This is because, for each timestep, only one pair of bias and RPReLU are selected for use.

## 4 Experiments

### 4.1 Experimental Settings

**Data and Evaluation.** We take DIV2K [59] and Flickr2K [33] as the training dataset. Meanwhile, we evaluate the models with four benchmark datasets: Set5 [2], B100 [42], Urban100 [18], and Manga109 [43]. Experiments are conducted under two upscale factors: ×2 and ×4. The LR images are generated from HR images through bicubic downsampling degradation. We apply two distortion-based metrics, PSNR and SSIM [64], which are calculated on the Y channel (*i.e.*, luminance) of the YCbCr space. We also use the perceptual metrics: LPIPS [12]. Following previous work [66, 49], the total parameters (**Params**) of the model are calculated as Params=Params$^b$+Params$^f$, and the overall operations (**OPs**) as OPs=OPs$^b$+OPs$^f$, where Params$^b$=Params$^f$/32 and OPs$^b$=OPs$^f$/64; the superscripts $f$ and $b$ denote full-precision and binarized modules, respectively.

**Implementation Details.** For the noise estimation network, we set the encoder and decoder level to 4. In each level of the encoder, we use 2 Residual Blocks (ResBlocks), while in the decoder, we apply 3 ResBlocks. The number of channels $C$ is set to 64. We set the number of bias and RPReLU in TaR and TaA as $K$=5. For the diffusion model, we set the total number of timesteps to $T$=2,000. During the inference phase, we employ the DDIM sampler with 50 timesteps.

**Training Settings.** We train models with the $\mathcal{L}_1$ loss. We employ the Adam optimizer [22] with $\beta_1$=0.9 and $\beta_2$=0.99, and a learning rate of $1\times10^{-4}$. The batch size is set to 16, with a total of 1,000K iterations. Input LR images are randomly cropped to size 64×64. Random rotations of 90°, 180°, and 270° and horizontal flips are used for data augmentation. Our model is implemented based on PyTorch [47] with two Nvidia A100-80G GPUs.

### 4.2 Ablation Study

In this section, we conduct all experiments on the ×2 scale factor. We apply DIV2K [59] and Flickr2K [33] as the training dataset, and Manga109 [43] as the testing dataset. The training iterations are set to 500K. Other settings are the same as defined in Sec. 4.1. We test the computational complexity (*i.e.*, OPs) of one single sampling step on the output size 3×256×256.

| Method | Baseline | +Identity | +CP-Down&Up | +CS-Fusion | +TaR&TaA |
|---|---|---|---|---|---|
| Params (M) | 4.29 | 4.29 | 4.29 | 4.30 | 4.58 |
| OPs (G) | 36.67 | 36.67 | 36.67 | 36.67 | 36.67 |
| PSNR (dB) | 27.66 | 29.29 | 31.08 | 31.99 | 32.66 |
| LPIPS | 0.0780 | 0.0658 | 0.0327 | 0.0261 | 0.0200 |

(a) Break-down ablation.

| Method | Params (M) | OPs (G) | PSNR (dB) | LPIPS |
|---|---|---|---|---|
| Add | 4.10 | 33.40 | 18.89 | 0.1695 |
| Concat | 4.29 | 36.67 | 31.08 | 0.0327 |
| Split | 4.30 | 36.67 | 29.67 | 0.0384 |
| CS-Fusion | 4.30 | 36.67 | 31.99 | 0.0261 |

(b) Ablation on feature fusion.

| Method | TaR | TaA | Params (M) | Ops (G) | PSNR (dB) | LPIPS |
|---|---|---|---|---|---|---|
| w/o | | | 4.30 | 36.67 | 31.99 | 0.0261 |
| In | ✓ | | 4.37 | 36.67 | 29.27 | 0.0337 |
| Out | | ✓ | 4.51 | 36.67 | 29.13 | 0.0308 |
| All | ✓ | ✓ | 4.58 | 36.67 | 32.66 | 0.0200 |

(c) Ablation on time aware module (TaR and TaA).

| #Pair | 1 | 2 | 5 |
|---|---|---|---|
| Params (M) | 4.30 | 4.37 | 4.58 |
| OPs (G) | 36.67 | 36.67 | 36.67 |
| PSNR (dB) | 31.99 | 32.42 | 32.66 |
| LPIPS | 0.0261 | 0.0229 | 0.0200 |

(d) Numbers (#) of bias and RPReLU pair.

Table 1: Ablation study. We train models on DIV2K and Flickr2K, and evaluate on Manga109 ($\times 2$).

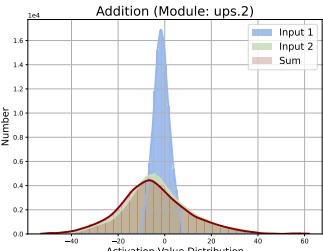 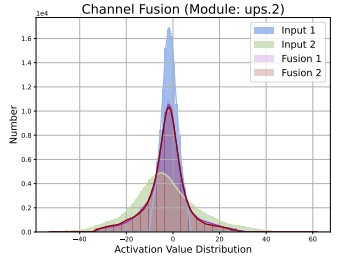 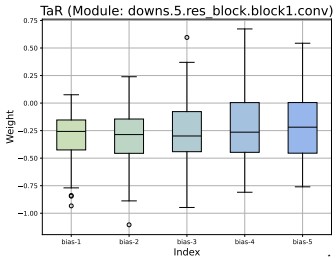

Figure 6: Activation distribution in the skip connection. Input 1(2): $\mathbf{x}_1$, $\mathbf{x}_2$. Sum: $\mathbf{x}_1 + \mathbf{x}_2$. Fusion 1(2): $\mathbf{x}_1^{sh}$, $\mathbf{x}_2^{sh}$.

Figure 7: Weights of biases $\mathbf{b}^i$ ($i \in \{1, \ldots, 5\}$) in TaR.

**Break Down.** We first execute a break-down ablation on different components of our method. The results are listed in Tab. 1a. The baseline is established by using binarized convolution (BI-Conv) and Pixel-(Un)Shuffle for dimension scaling in the downsample, upsample, and fusion (skip connection) modules of the UNet. Meanwhile, the basic BI-Conv block (Fig. 5) is employed without the identity shortcut. The baseline performance is poor, with the PSNR of 27.66 dB. Then, we add identity shortcut, consistent-pixel-downsample (CP-Down) and upsample (CP-Up), channel-shuffle-fusion module (CS-Fusion), and timestep-aware redistribution (TaR) and activation function (TaA) in sequence. We can find that the performance gradually increases. Ultimately, the final model achieves gains of 5 dB in PSNR and 0.0580 in LPIPS, compared to the baseline.

**Channel-Shuffle Fusion.** We experiment on the fusion module for the skip connection. We attempt four methods: directly add two features (Add); concatenation and adjust dimension by binarized convolution (Concat); process each feature via binarized convolution and add them; and our proposed CS-Fusion. The results are shown in Tab. 1b. Due to the differences between features, direct addition (Add) can hardly work, even with convolution (Split). Moreover, since the concatenation changes the dimensions, the Method (Concat) also degrades the performance. In contrast, our proposed CS-Fusion, eliminates the distribution imbalances by channel fusion, thereby achieving effective fusion. The visualization in Fig. 6, further indicates that addition cannot fuse data with narrow value distributions, whereas channel shuffle can effectively integrate.

**Timestep-aware Module.** We conduct experiments on the time-aware redistribution (TaR) and activation function (TaA). Firstly, we experiment with the combinations of TaR and TaA in Tab. 1c. We find that effective improvements are only achieved when both TaR and TaA are employed. This may be because both input and output activation impact the learning of the binarized module. Then, in Tab. 1d, we experiment with the pair number (#Pair) of bias and RPReLU. The experiments show that 5 pairs already lead to effective improvements. Considering the additional parameters, we adopt 5 as the pair number in BI-DiffSR. Moreover, we present the weights of five learnable biases in the TaR (module position shown at the image top) in Fig. 7. The difference in weights indicates that TaR can effectively adapt to the varying activation distributions at different timesteps.

### 4.3 Comparison with State-of-the-Art Methods

We compare our proposed BI-DiffSR with recent binarization methods, including BNN [19], DoReFa [71], XNOR [50], IRNet [48], ReActNet [38], and BBCU [66]. To ensure a fair comparison, we set the parameters (Params) and complexity (OPs) of all binarization methods to be similar. We also compare our BI-DiffSR with the full-precision (FP) model, SR3 [54].

| Method | Scale | Params (M) | Ops (G) | Set5 PSNR | Set5 SSIM | Set5 LPIPS | B100 PSNR | B100 SSIM | B100 LPIPS | Urban100 PSNR | Urban100 SSIM | Urban100 LPIPS | Manga109 PSNR | Manga109 SSIM | Manga109 LPIPS |
|---|---|---|---|---|---|---|---|---|---|---|---|---|---|---|---|
| Bicubic | ×2 | N/A | N/A | 33.67 | 0.9303 | 0.1274 | 29.55 | 0.8431 | 0.2508 | 26.87 | 0.8403 | 0.2064 | 30.82 | 0.9349 | 0.1025 |
| SR3 [54] | ×2 | 55.41 | 176.41 | 36.69 | 0.9513 | 0.0310 | 30.41 | 0.8683 | 0.0700 | 30.29 | 0.9060 | 0.0430 | 35.11 | 0.9682 | 0.0161 |
| BNN [19] | ×2 | 4.78 | 37.93 | 13.97 | 0.5210 | 0.4529 | 13.73 | 0.4553 | 0.5784 | 12.75 | 0.4236 | 0.5575 | 9.29 | 0.3035 | 0.7489 |
| DoReFa [71] | ×2 | 4.78 | 37.93 | 16.43 | 0.6553 | 0.2662 | 16.11 | 0.5912 | 0.3972 | 15.09 | 0.5495 | 0.4055 | 12.35 | 0.4609 | 0.5047 |
| XNOR [50] | ×2 | 4.78 | 37.93 | 32.34 | 0.8661 | 0.0782 | 27.94 | 0.7548 | 0.1665 | 27.47 | 0.8225 | 0.1153 | 31.99 | 0.9428 | 0.0326 |
| IRNet [48] | ×2 | 4.78 | 37.93 | 32.55 | 0.9340 | 0.0446 | 27.76 | 0.8199 | 0.1115 | 26.34 | 0.8452 | 0.0913 | 23.89 | 0.7621 | 0.1820 |
| ReActNet [38] | ×2 | 4.85 | 37.93 | 34.30 | 0.9271 | 0.0351 | 28.36 | 0.8158 | 0.0943 | 27.43 | 0.8563 | 0.0731 | 32.16 | 0.9441 | 0.0379 |
| BBCU [66] | ×2 | 4.82 | 37.75 | 34.31 | 0.9281 | 0.0393 | 28.39 | 0.8202 | 0.0905 | 28.05 | 0.8669 | 0.0620 | 32.88 | 0.9508 | 0.0272 |
| BI-DiffSR (ours) | ×2 | 4.58 | 36.67 | 35.68 | 0.9414 | 0.0277 | 29.73 | 0.8478 | 0.0682 | 28.97 | 0.8815 | 0.0522 | 33.99 | 0.9601 | 0.0172 |
| Bicubic | ×4 | N/A | N/A | 28.43 | 0.8111 | 0.3398 | 25.95 | 0.6678 | 0.5244 | 23.14 | 0.6579 | 0.4729 | 24.90 | 0.7876 | 0.3210 |
| SR3 [54] | ×4 | 55.41 | 176.41 | 31.03 | 0.8798 | 0.1127 | 26.11 | 0.6933 | 0.2247 | 25.52 | 0.7702 | 0.1438 | 28.77 | 0.8854 | 0.0646 |
| BNN [19] | ×4 | 4.78 | 37.93 | 12.21 | 0.3103 | 0.8310 | 12.30 | 0.2128 | 0.9519 | 11.30 | 0.2191 | 0.9592 | 8.96 | 0.1833 | 1.0117 |
| DoReFa [71] | ×4 | 4.78 | 37.93 | 10.40 | 0.246 | 0.9855 | 9.78 | 0.1709 | 1.0793 | 8.79 | 0.1614 | 1.1186 | 7.52 | 0.1464 | 1.1169 |
| XNOR [50] | ×4 | 4.78 | 37.93 | 28.06 | 0.8274 | 0.1381 | 25.25 | 0.6552 | 0.3101 | 23.13 | 0.6647 | 0.2564 | 23.84 | 0.7839 | 0.1559 |
| IRNet [48] | ×4 | 4.78 | 37.93 | 15.52 | 0.3514 | 0.7548 | 16.38 | 0.3121 | 0.7072 | 15.23 | 0.3043 | 0.7068 | 11.82 | 0.2442 | 0.8354 |
| ReActNet [38] | ×4 | 4.85 | 37.93 | 29.23 | 0.8362 | 0.1472 | 23.56 | 0.5670 | 0.3339 | 22.32 | 0.6440 | 0.2276 | 25.32 | 0.7854 | 0.1721 |
| BBCU [66] | ×4 | 4.82 | 37.75 | 25.44 | 0.7795 | 0.1650 | 21.46 | 0.5472 | 0.3206 | 20.52 | 0.6293 | 0.2290 | 23.02 | 0.7966 | 0.1496 |
| BI-DiffSR (ours) | ×4 | 4.58 | 36.67 | 29.63 | 0.8374 | 0.1109 | 25.84 | 0.6779 | 0.2754 | 24.11 | 0.7177 | 0.1823 | 26.95 | 0.8548 | 0.0889 |

Table 2: Quantitative comparison with state-of-the-art binarization methods. The best and second best results are coloured with red and blue. Our method surpasses current approaches.

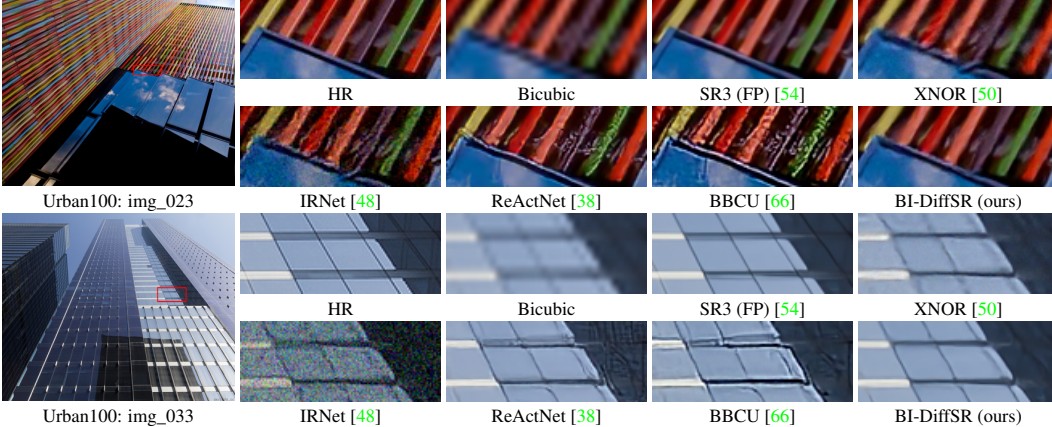

HR | Bicubic | SR3 (FP) [54] | XNOR [50]

Urban100: img_023 | IRNet [48] | ReActNet [38] | BBCU [66] | BI-DiffSR (ours)

HR | Bicubic | SR3 (FP) [54] | XNOR [50]

Urban100: img_033 | IRNet [48] | ReActNet [38] | BBCU [66] | BI-DiffSR (ours)

Figure 8: Visual comparison (×4) in some challenge cases.

**Quantitative Results.** We provide the quantitative comparisons in Tab. 2. We test OPs of single-step sampling on the output size 3×256×256. Compared to other binarization methods, our BI-DiffSR achieves the best performance. Specifically, on Urban100 and Manga109 (×2), BI-DiffSR surpasses the second-best method, BBCU, with a PSNR gain of **0.92** and **1.11** dB, respectively. Moreover, compared to the full-precision model, SR3, our method achieves comparable or even better perceptual performance with only 8.3% Params and 20.8% OPs. For instance, BI-DiffSR achieves 93.6% LPIPS results of SR3 on Manga109. These results demonstrate the superiority of our method.

**Visual Results.** We present visual comparisons (×4) in Fig. 8. Previous binarization methods struggle to recover image details in challenging cases. In contrast, our method can restore clearer results with more texture details. Meanwhile, the difference between our BI-DiffSR and the full-precision model results is small. More visual results are provided in the supplementary material.

## 5 Conclusion

In this paper, we propose the BI-DiffSR, a novel binarized diffusion model for image SR. Specifically, we first design the UNet structure suitable for binarization. To ensure dimension consistency and full-precision information transfer, we design the consistent-pixel-downsample (CP-Down) and upsample (CP-Up). Meanwhile, we develop the channel-shuffle-fusion (CS-Fusion) to enhance information fusion within the skip connection. Furthermore, in response to the multi-step mechanism of diffusion models, we design the timestep-aware redistribution (TaR) and activation functions (TaA) to adapt to the varying activation distributions. The TaR and TaA enhance the representational capabilities of the binarized modules under multiple timesteps. Extensive experiments indicate that our method outperforms current binarization methods, and achieves comparable perceptual performance to the full-precision model, demonstrating substantial potential.

**Acknowledgments**. This work is supported by the National Natural Science Foundation of China (62141220, 62271414), Shanghai Municipal Science and Technology Major Project (2021SHZDZX0102), the Fundamental Research Funds for the Central Universities, Zhejiang Provincial Distinguish Young Science Foundation (LR23F010001), Zhejiang "Pioneer" and "Leading Goose" R&D Program (2024SDXHDX0006, 2024C03182), the Key Project of Westlake Institute for Optoelectronics (2023GD007), and Ningbo Science and Technology Bureau, "Science and Technology Yongjiang 2035" Key Technology Breakthrough Program (2024Z126).

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
