# OpenReview forum: "Binarized Diffusion Model for Image Super-Resolution"
_NeurIPS.cc/2024/Conference — NeurIPS 2024 poster_

### Official Review · Reviewer_MRyP · 2024-06-28

**Soundness:** 4
**Presentation:** 4
**Contribution:** 4
**Rating:** 6
**Confidence:** 4

**Summary:**

The paper introduces BI-DiffSR, a novel binarized diffusion model for image super-resolution, designed to accelerate the inference speed and reduce computational costs of diffusion models while maintaining high performance. It proposes a UNet architecture optimized for binarization, featuring consistent-pixel downsampling/upsampling and channel-shuffle fusion to address dimension mismatch and fusion difficulty, alongside a timestep-aware redistribution and activation function to adapt to varying activation distributions across different timesteps. The model demonstrates superior results over existing binarization methods, approaching the perceptual quality of full-precision models with significantly reduced memory and computational requirements.

**Strengths:**

- The paper is well-written and easy to understand.

- This paper designs a novelty 1-bit UNet for accurate binarized diffusion model, including:
   - New downsample module and upsample module for Dimension Consistency.
   - Channel shuffle module to balance the activation value ranges of two input features.
   - The timestep-aware redistribution (TaR) and timestep-aware activation function (TaA)

- Experiments achieve the state-of-the-art in super resolution with diffusion.

**Weaknesses:**

- The basic BI-Conv block lacks novelty, which is as the same as the binarized module in ReActNet that contains RSign and RPReLU.

- TaR uses different parameters for different time steps, but in the mean while, the normal time embedding is projected into the resblock, it is also a time-aware on feature maps, what is the differences or why TaR works?

- SR3 is not a new diffusion baseline for super resolution, ResShift[1], SinSR[2] should be better, and the metrics of PSNR, SSIM, LPIPS is much old, the CLIPIQA, MUSIQ, MANIQA should be better for evaluating the performance of generative super resolution.

- Self-attention and MLP are common modules in diffusion, such as LDM[3] and ResShift[1], which require a lot of computation. How can the method in this paper be extended to self-attention and MLP?


[1] Yue, Zongsheng, Jianyi Wang, and Chen Change Loy. "Resshift: Efficient diffusion model for image super-resolution by residual shifting." Advances in Neural Information Processing Systems 36 (2024).

[2] Wang, Yufei, et al. "SinSR: diffusion-based image super-resolution in a single step." Proceedings of the IEEE/CVF Conference on Computer Vision and Pattern Recognition. 2024.

[3] Rombach, Robin, et al. "High-resolution image synthesis with latent diffusion models." Proceedings of the IEEE/CVF conference on computer vision and pattern recognition. 2022.

**Questions:**

Please refer to the weaknesses above.

**Limitations:**

The authors have addressed the limitations in the paper.

---

> ### Author Rebuttal · Authors · 2024-08-06
>
> # Response to Reviewer MRyP (denoted as R4)
>
>
>
> `Q4-1` The basic BI-Conv block lacks novelty, which is as the same as the binarized module in ReActNet that contains RSign and RPReLU.
>
> `A4-1` Thanks for pointing it out. We clarify it below.
>
> 1. Indeed, our basic BI-Conv block utilizes RSign and RPReLU for the learnable bias and activation function. However, this is **not** an innovative aspect of our method.
>
> 2. Our **innovation** lies in timestep-aware operations (*i.e.*, TaR and TaA). Inspired by the mixture of experts (MoE), we employ different RSign and RPReLU according to different timesteps.
>
> 3. We demonstrate in **Table 1c** of the main paper that using timestep-aware operations effectively improves the performance of binarized DM.
>
>    | Method       | Params (M) | OPs (G) | PSNR (dB) | LPIPS  |
>    | :----------- | :--------: | :-----: | :-------: | :----: |
>    | RSign&RPReLU |    4.30    |  36.67  |   31.99   | 0.0261 |
>    | TaA&TaR      |    4.58    |  36.67  |   32.66   | 0.0200 |
>
> 4. To enhance transparency, we will clarify in the manuscript that RSign and RPReLU are from ReActNet.
>
>
>
> `Q4-2` TaR uses different parameters for different time steps, but in the mean while, the normal time embedding is projected into the resblock, it is also a time-aware on feature maps, what is the differences or why TaR works?
>
> `A4-2` We explain the differences below.
>
> **Differences:**
>
> 1. **Location:** Time embedding acts on the **ResBlock**; TaR operates within the **BI-Conv** (inside the ResBlock).
> 2. **Purpose:** Time embedding operates at **the network level**, enabling the model to be aware of different timesteps and enhance feature modeling; TaR operates at **the module level** (such as Conv), adjusting the activation distribution based on the timestep to improve module computations.
>
> **Functionality of TaR:**
>
> 1. **Adapting Activation Distribution:** TaR dynamically adjusts Conv input activation according to different timesteps, adapting changing distributions.
> 2. **Enhancing Conv Representation:** TaR divides multiple steps into smaller groups, limiting the changing range of activations, thereby reducing the representational difficulty of BI-Conv and enhancing feature extraction.
> 3. **Complementarity with Time Embedding:** TaR and time embedding serve different purposes and are not contradictory. Using TaR (along with TaA) in DM further enhances model performance. This is evidenced in **Table 1c** (as detailed in `A4-1`).
>
>
>
> `Q4-3` SR3 is not a new diffusion baseline for super resolution, ResShift[1], SinSR[2] should be better, and the metrics of PSNR, SSIM, LPIPS is much old, the CLIPIQA, MUSIQ, MANIQA should be better for evaluating the performance of generative super resolution.
>
> `A4-3` Thanks for your suggestion. We apply the new baseline **ResShift** [1] (official code), and compare our method, BI-DiffSR, with the full-precision (FP) model and the previous binarized model, BBCU.
>
> **Setup:**
>
> 1. We train on the bicubic-×4 task on the DF2K dataset, batch size 32, iteration 400,000, sample step 15, maintaining other settings consistent with the official setup.
> 2. As our method involves adjusting the channels of UNet, we modify the UNet structure (maintaining the core module unchanged) to create **ResShift\***. We then apply **BBCU** and **BI-DiffSR** (quantizing both Conv and SA) on ResShift\* to ensure fairness.
> 3. We measure the FLOPs on an input size of 128×128. All metrics are evaluated on Manga109.
>
> | Method                      | Params (M) | OPs (G) | CLIPIQA $\uparrow$ | MANIQA $\uparrow$ | MUSIQ $\uparrow$ |
> | :-------------------------- | :--------: | :-----: | :----------------: | :---------------: | :--------------: |
> | ResShift* (full-precision)  |   58.12    | 170.77  |       0.7662       |      0.5543       |     67.3729      |
> | BBCU (binarized)            |    3.89    |  20.19  |       0.7611       |      0.5175       |     64.0867      |
> | BI-DiffSR (binarized, ours) |    3.65    |  18.57  |       0.7659       |      0.5638       |     66.5730      |
>
> **Results Analyses:**
>
> 1. **Comparison with BBCU:** Our method performs better across all new metrics.
> 2. **Comparison with ResShift\*:** Our method achieves similar performance while significantly reducing parameters (**93.72%**) and operations (**89.13%**).
> 3. **General Effectiveness:** These results further demonstrate the generalizability and effectiveness of our method.
>
>
>
> **Further Consideration:**
>
> 1. **Baseline Choice:** In our paper, we use **SR3** as the baseline, considering it is a normal DM SR method, which can demonstrate the generality of our approach. Classic metrics such as PSNR, SSIM, and LPIPS are used for similar reasons.
> 2. **Inclusion of New Baseline:** As the reviewer suggested, using new baselines and metrics better reflects the effect of our method. We will include ResShift-based experiments in our paper for a thorough assessment.
>
>
>
> `Q4-4` Self-attention and MLP are common modules in diffusion, such as LDM[3] and ResShift[1], which require a lot of computation. How can the method in this paper be extended to self-attention and MLP?
>
> `A4-4` The methods are below.
>
> **Extension to Self-Attention (SA):**
>
> 1. **Linear Layers:** SA involves three linear layers for $Q, K, V$, and a linear projection for the output. These parts can be replaced with 1×1 BI-Conv to achieve binarization.
> 2. **Matrix Operations:** The matrix operation, $\text{Attention}(Q, K, V) = \text{softmax}\left(\frac{QK^T}{\sqrt{d_k}}\right)V$, can be binarized by binarizing the activations: $Q, K, V, \text{softmax}\left(\frac{QK^T}{\sqrt{d_k}}\right)$, through the sign function, Sign(·).
>
> **Extension to MLP:**
>
> 1. **Linear Layers:** The MLP consists of several linear layers, which can be directly replaced with 1×1 BI-Conv for binarization.
>
> **Experiment:** We binarize **ResShift [1]** with our method, effectively reducing Params and OPs while maintaining performance, as detailed in `A4-3`.

---

> > ### Comment · Reviewer_MRyP · 2024-08-12
> > **Thank you for the rebuttal**
> >
> > Thank you for the response and additional experiments.BI-DiffSR is insteresting and  promising to push the development of  diffusion deployment.  After reviewing the rebuttal, I decided to raise my score.

---

> > > ### Author Response · Authors · 2024-08-12
> > > **Thanks Reviewer MRyP for approving our work**
> > >
> > > Dear Reviewer MRyP,
> > >
> > > Thank you for your response. We are pleased that you approve of our work.
> > >
> > > Best,
> > > Authors

---

### Official Review · Reviewer_KWS7 · 2024-07-08

**Soundness:** 4
**Presentation:** 3
**Contribution:** 4
**Rating:** 8
**Confidence:** 4

**Summary:**

This work present a novel binarized diffusion model for improving the efficiency of super resolution tasks. Compared with the existing works, this work first pointed out the specific challenges of binarized DMs for SR, including the dimension mismatch and fusion difficulty of representations. Then this work present several techniques: consistent-pixel down/upsampleing, channel-shuffle fusion, and Time-step-aware redistribution function for the aforementioned challenges. Comprehensive results show that the provided binarized DMs for SR not only significantly outperform the binarized models with existing SOTA binarization methods, but also achieve floating-point level performance. And for the efficiency, the statistics of params and flops show the advantage of proposed method, and the paper also present the real inference time on edge, which seems important and encouraged in the binarization community.

**Strengths:**

1. As far as I know, this is the first work to present the specific binarization method for diffusion model of SR. Since the good performance has been achieved by DMs in various SR tasks, it’s important to present novel insight to compress these models, especially considering the severe drop still exists after binarizing by existing SOTA methods.
2. The motivation is intuitive and techniques are novelty, especially considering the features of DMs. The proposed CP-Up/down and channel shuffle are highly specified to the architecture of the diffusion models, which is novel and cannot be achieved by previous methods, including binarization function and binarized structures. And the computation is also small, allowing minor burden with significant performance improvement. And the proposed activation function also focus on the high dynamic of activation range during  time-step, which is one of the most critical problem for the quantization of DMs.
3. The proposed method achieve SOTA results in accuracy. Comprehensive comparison has been included in this paper, including SOTA binarization methods and various evaluation datasets. The results show that the proposed outperforms than previous binarized DMs for SR with significant improvements.
4. In this paper, diverse analysis, including quantitative, statistical, and visual results are presented in detail. More important, the paper shows the efficiency evaluation based on real inference libraries and edge hardware, which is of great significance for practical application.

**Weaknesses:**

Though it’s a good paper, some issues should be addressed.
1. The writing and presentation of the paper should be improved, including but not limited to the grammar and description. For example, some basic knowledge about quantization, SR, and DMs seems to be summarized as a preliminaries section; and let the proposed techniques be highlighted in Figure 2.
2. As for the efficiency, I suggest the authors present the computation more detailed, such as present the computation of each part in the whole network before and after the binarization. This will show the efficiency advantage of the proposed method much clearer.
3. The proposed challenge I and II are insightful, but more further discussion (such as visual, quantitative, or theoretical analysis) are presented after proposing. I suggest authors do more discussion about that.
4. Some recent binarization methods for SR [1] are suggested to be compared and some quantized DMs [2] are suggested to be discussed the differences to make the comparison more comprehensive.
[1] Flexible Residual Binarization for Image Super-Resolution. ICML 2024
[2] EfficientDM: Efficient Quantization-Aware Fine-Tuning of Low-Bit Diffusion Models. ICLR 2024

**Questions:**

1. Compared with the quantization of DMs for SR, can authors provide discussion about the advantage and motivation of binarization?
2. What is the type of ARM hardware for the evaluation of inference?
3. If the proposed method have potential generalized to more generative tasks and architectures?

**Limitations:**

The authors have addressed the limitations.

---

> ### Author Rebuttal · Authors · 2024-08-06
>
> # Response to Reviewer KWS7 (denoted as R3)
>
>
>
> `Q3-1` The writing and presentation of the paper should be improved, including but not limited to the grammar and description. For example, some basic knowledge about quantization, SR, and DMs seems to be summarized as a preliminaries section; and let the proposed techniques be highlighted in Figure 2.
>
> `A3-1` Thank you for your suggestions. We will check and improve our manuscript.
>
> We have added a **preliminaries section**:
>
> >**SR Pipeline.** SR network aims to reconstruct a low-resolution image into a corresponding high-resolution image. The process can be represented as follows:
> >$$
> >I_{SR} = \mathcal{SR} (I_{LR} ; \Theta),
> >$$
> >where $\mathcal{SR} (\cdot)$ denotes the image SR network, and $\Theta$​ represents the network parameters.
> >
> >**Binarization Framework.** In binarized networks, weights and activations are converted using the sign function:
> >$$
> >\operatorname{Sign}\left(x\right)= \begin{cases}+1, & x \geq 0 \\\ -1, & x < 0 \end{cases}.
> >$$
> >As the sign function Sign(⋅) is non-differentiable, we use the straight-through estimator (STE) for backpropagation to train binarized models:
> >$$
> >\frac{\partial \operatorname{sign}}{\partial \boldsymbol{x}}= \begin{cases}1 & \text { if }|\boldsymbol{x}| \leq 1 \\\ 0 & \text { otherwise }\end{cases}.
> >$$
> >Binarization reduces storage and accelerates computation through XNOR and bit-counting operations.
>
> For **Figure 2**, we have modified the image to highlight our proposed techniques and added it to the **attached PDF (Figure 2)**.
>
>
>
> `Q3-2` As for the efficiency, I suggest the authors present the computation more detailed, such as present the computation of each part in the whole network before and after the binarization. This will show the efficiency advantage of the proposed method much clearer.
>
> `A3-2` Thank you for your suggestion. We present the Params and OPs for each part of the UNet network: encoder, bottleneck, and decoder, before and after binarization. OPs are tested with an output of 3×256×256.
>
> | Part       | Params$^f$ (M) | Params$^d$ (M) | OPs$^f$ (G) | OPs$^d$ (G) |
> | ---------- | :------------: | :------------: | :---------: | :---------: |
> | Encoder    |     13.09      |      0.65      |    42.66    |    1.04     |
> | Bottleneck |     10.59      |      1.45      |    13.97    |    2.31     |
> | Decoder    |     27.30      |      2.46      |    79.29    |    33.31    |
>
> 1. Params$^f$ and OPs$^f$ correspond to the full-precision modules, while Params$^d$ and OPs$^d$ are for the binarized versions.
>
> 2. The encoder and bottleneck show high compression ratios. To balance parameters and performance, we use a partial number full-precision ResBlocks in the decoder, which reduces the compression ratio.
>
>
>
> `Q3-3` The proposed challenge I and II are insightful, but more further discussion (such as visual, quantitative, or theoretical analysis) are presented after proposing. I suggest authors do more discussion about that.
>
> `A3-3` Thank you for your suggestion. We add more visual results for the two challenges in the **attached PDF (Figure 3)**. Here are the analyses:
>
> **Challenge I: Dimension Mismatch.** The binarized modules struggle with detailed features. Using full-precision features through residual connections helps, as shown in columns three and four of Figure 3. However, in UNet, up/down sampling makes feature shape mismatches, making it unable to use residual connections. We solve this problem via CP-Down/Up.
>
> **Challenge II: Fusion Difficulty.** The activation distributions of skip connections vary greatly, resulting in information loss in the decoding stage and a lack of details in the restored image. Therefore, we design CS-Fusion to address this problem, as shown in columns four and five of Figure 3.
>
>
>
> `Q3-4` Some recent binarization methods for SR [1] are suggested to be compared and some quantized DMs [2] are suggested to be discussed the differences to make the comparison more comprehensive.
>
> `A3-4` Thanks for your advice. We discuss the differences with FRB [1] and EfficientDM [2] below.
>
> **FRB [1]**
>
> 1. **Network Design:** FRB targets one-step networks; our method includes dimension and timestep considerations in DMs.
> 2. **Training Approach:** FRB uses full-precision distillation to guide the training; we use a basic L1 loss to train our model.
>
> **EfficientDM [2]**
>
> 1. **Quantization Level:** EfficientDM implements low-bit (2/4/8) quantization; we consider the extreme case of 1-bit (binarization).
> 2. **Activation Management:** EfficientDM employs temporal activation LSQ to handle changes in activation distribution; we propose timestep-aware techniques (TaA and TaR).
>
>
>
> `Q3-5` Compared with the quantization of DMs for SR, can authors provide discussion about the advantage and motivation of binarization?
>
> `A3-5` We discuss them below.
>
> **Advantage:**
>
> 1. **Higher Compression Ratio:** Binarization (1-bit) offers the highest parameter reduction compared to 2/4/8-bit quantizations.
> 2. **Efficient Computation:** Binarization allows the model to perform inference via bitwise operations, a capability not inherent to other bit-level models.
>
> **Motivation:** Diffusion models (DMs) have excellent generation abilities but are resource-intensive. Binarization greatly reduces this overhead, enhancing usability on limited-capacity devices.
>
>
>
> `Q3-6` What is the type of ARM hardware for the evaluation of inference?
>
> `A3-6` The evaluation is conducted on a **Raspberry Pi 3 Model B+** (BCM2837B0, Cortex-A53 (ARMv8) 64-bit SoC @ 1.4GHz).
>
>
>
> `Q3-7` If the proposed method have potential generalized to more generative tasks and architectures?
>
> `A3-7` Yes, it has the potential.
>
> 1. **Other Tasks:** The model can be directly applied to unconditional generative tasks by removing the LR input.
> 2. **Other Architectures:** Many generative diffusion models, like stable diffusion, can be binarized by our proposed approach.

---

> > ### Comment · Reviewer_KWS7 · 2024-08-12
> > **After Rebuttal**
> >
> > Thank you to the authors for their detailed response. After reviewing all the explanations, I can confirm that all of my concerns have been fully addressed. I would like to keep my original score.

---

> > > ### Author Response · Authors · 2024-08-13
> > > **Thanks Reviewer KWS7 for approving our work**
> > >
> > > Dear Reviewer KWS7,
> > >
> > > Thank you for your response. We are honoured that our replies have addressed the reviewer's concerns. We sincerely appreciate your thorough review and valuable suggestions.
> > >
> > > Best,
> > > Authors

---

### Official Review · Reviewer_Z33c · 2024-07-09

**Soundness:** 3
**Presentation:** 4
**Contribution:** 3
**Rating:** 7
**Confidence:** 4

**Summary:**

The authors propose BI-DiffSR to binarize diffusion based image super-resolution (SR) model. They design a UNet architecture for the whole binarized model structure. To maintain dimension consistency, they propose two modules, CP-Down and CP-Up, which can further help transfer full-precision information. To enhance feature fusion, they propose the channel-shuffle-fusion (CS-Fusion). They also propose TaR and TaA to dynamically adjust activation distribution cross different timesteps. The authors provide extensice experiments to demonstrate the effectiveness of their proposed method.

**Strengths:**

The topic is very important and practical. Diffusion models have shown excellent performance for image super-resolution (SR). It is very practical to quantize the models before deploying them into devices. Binarization is an extreme tool to compress the SR model. Few works have been proposed to investigate such an important problem in image SR.

The authors give several insights for the specific topic. Namely, there are some key aspects in diffusion based image SR binarization, like dimension mismatch, fusion difficulty, and activation distraction. Those problems hinder the performance of binarized image SR diffusion models. The observation and analyses given in the introduction section are insightful and motivate readers well.

To alliveate the problems in binarized diffuision based SR models, the authors propose consistent-pixel-downsample (CP-Down) and consistent-pixel-upsample (CP-Up) to ensure dimensional consistency. They propose the channel-shuffle-fusion (CS-Fusion) to facilitate the fusion of different features within skip connections and suit binarized modules. They propose the timestep-aware redistribution (TaR) and timestep-aware activation function (TaA) to adjust the binarized module input and output arross different timesteps.

They provide extensive ablation study experiments (including quantitative results in Table 1 and visualization analyses in Figures 6 and 7.) to show the effects of each proposed components. Those experiments are convincing.

The authors provide comparions with SOTA methods. According to the main quantitive and visual comparisons, they show that their proposed BI-DiffSR achieves superior performance over others.

The overall writing and organization are pretty good. I think the work is well-prepared. The supplementary file further provides more details. The paper is easy to follow and they promise to release the code, which makes this work more convincing.

**Weaknesses:**

When binarizing full-precision model from 32-bit to 1-bit, ideally we can reduce the parameters by 32 times. But, as shown in Table 2, the authors reduce parameters from 55.41M to 4.58M (for scale 2). There is a gap between ideal case and practical one. Please give some analyese about the reasons for this gap. Also, are there any idea to further narrow the gap?

The parameters and Ops are reduced obviously from full-precision to binary one. But the authors did not give results about inference time on real devices or give some analyses. I am curious how fast the binarized model will be.

The writing can further refine in some cases. For example, in the abstract part (Line 9-10), “… to maintain dimension consistent” should be changed to “… to maintain dimension consistency”.

**Questions:**

Can this method be applied to other diffusion models, like stable diffusion? If so, can the authors give some suggestions to binarize stable diffusion?

Can we apply this binarization method to other related image restoration tasks? Like image denoising, deblurring?

How long did the authors to train the models?

**Limitations:**

Please refer to weaknesses

---

> ### Author Rebuttal · Authors · 2024-08-06
>
> # Response to Reviewer Z33c (denoted as R2)
>
>
>
> `Q2-1` When binarizing full-precision model from 32-bit to 1-bit, ideally we can reduce the parameters by 32 times. But, as shown in Table 2, the authors reduce parameters from 55.41M to 4.58M (for scale 2). There is a gap between ideal case and practical one. Please give some analyese about the reasons for this gap. Also, are there any idea to further narrow the gap?
>
> `A2-1` **Reasons:**
>
> 1. We use a partial number (*i.e.*, 6) full-precision (FP) ResBlocks in the UNet decoder part to trade off parameters and performance.
> 2. The proposed timestep-aware redistribution (TaR) and activation function (TaA) add ~0.3M parameters.
>
>
>
> **Strategy to Narrow the Gap:** In practice, we can adjust the number of FP ResBlocks according to the situation to suit different environments. For instance, on platforms with significant resource constraints, using fewer FP ResBlocks (and correspondingly more binarized ResBlocks) to reduce Params is acceptable, even if it slightly lowers performance.
>
>
>
> `Q2-2` The parameters and Ops are reduced obviously from full-precision to binary one. But the authors did not give results about inference time on real devices or give some analyses. I am curious how fast the binarized model will be.
>
> `A2-2` Thank you for your suggestion. In **Supplementary Material A.2**, we have compared the inference time of SR3 (FP) and BI-DiffSR (ours).
>
> | Method           | Params (M) | OPs (G) | Simulated Time (s) |
> | ---------------- | :--------: | :-----: | :----------------: |
> | SR3              |   55.41    | 176.41  |       55.37        |
> | BI-DiffSR (ours) |    4.58    |  36.67  |       13.00        |
>
> Our method operates faster compared to the full-precision approach (SR3).
>
> **Note:** Real device testing for binary models is limited by specific hardware requirements. Hence, we estimate inference times using the daBNN inference frame. The correlation between the operation count and running time on an **ARM64 CPU** is: the FP module costs 313.85 ms per GOP, and the BI module costs 354.56 ms per GOP.
>
>
>
> `Q2-3` The writing can further refine in some cases. For example, in the abstract part (Line 9-10), “… to maintain dimension consistent” should be changed to “… to maintain dimension consistency”.
>
> `A2-3` Thank you for the suggestion. We will check the entire paper and improve the writing.
>
>
>
> `Q2-4` Can this method be applied to other diffusion models, like stable diffusion? If so, can the authors give some suggestions to binarize stable diffusion?
>
> `A2-4` Other diffusion models, such as stable diffusion (SD), can be binarized by our method. The SD also employs the UNet noise estimation network, which is composed of ResBlocks and AttentionBlocks.
>
> **Methodology:**
>
> 1. **Dimension Matching:** Adjust the number of channels in different layers of the UNet to match dimensions via our proposed CP-Down, CP-Up, and CS-Fusion.
> 2. **ResBlock Binarization:** Directly binarize it using our proposed binary convolution (BI-Conv).
> 3. **AttentionBlock Binarization:** For this block, the core is to binarize self-attention (**SA**). This can be extended from BI-Conv. The SA obtains $Q, K, V$ by three linear projections, and gets the output by another linear. We can use 1x1 BI-Conv to binarize the linear layer. And for the attention computation $\text{Attention}(Q, K, V) = \text{softmax}\left(\frac{QK^T}{\sqrt{d_k}}\right)V$, the activations $Q, K, V$, and the intermediate attention map $\text{softmax}\left(\frac{QK^T}{\sqrt{d_k}}\right)$ can be binarized by the sign function Sign(·).
>
>
>
> `Q2-5` Can we apply this binarization method to other related image restoration tasks? Like image denoising, deblurring?
>
> `A2-5` Yes, our method can be applied to other tasks.
>
> 1. **Direct Application:** Our BI-DiffSR model can be directly trained on specific tasks like deblurring using datasets such as GoPro, without structural modifications.
> 2. **Binarization Task-Specific Model:** Our binarization method can also binarize diffusion models specific to these tasks.
>
>
>
> `Q2-6` How long did the authors to train the models?
>
> `A2-6` It takes approximately 84 hours to train the BI-DiffSR (×2) on two NVIDIA A100 GPUs.

---

> > ### Comment · Reviewer_Z33c · 2024-08-11
> >
> > I appreciate your thoughtful and detailed replies to my questions. My concerns have been well solved, thus, I tend to increase my score to 7.

---

> > > ### Author Response · Authors · 2024-08-11
> > > **Thanks Reviewer Z33c for approving our work**
> > >
> > > Dear Reviewer ymRm,
> > >
> > > Thanks for your response. We are happy to see that our answers can solve your concerns.
> > >
> > > Best,
> > > Authors

---

### Official Review · Reviewer_mEsa · 2024-07-11

**Soundness:** 3
**Presentation:** 4
**Contribution:** 2
**Rating:** 5
**Confidence:** 5

**Summary:**

This paper introduce a novel binarized diffusion model, BI-DiffSR, for image SR. A UNet architecture optimized for binarization, channel shuffle fusion, and time-step-aware redistribution and activation functions are designed. The experimental results proved the effectiveness of the method.

**Strengths:**

1. This paper is well written, nicely presented, and well organized.

2. Binarized diffusion networks are promising.

3. The performance improvement over other binary SR networks is significant.

**Weaknesses:**

1. Lack of discussion with some related works[1, 2, 3, 4], in particular [1] which is also for binary SR networks. Please analyze and discuss the differences with [1,2].

2. Ablation experiments are not convincing enough. Comparisons with some other activation function or fusion methods [1, 2, 3, 4] should be included.

3. It is always known that diffusion models are slow. Although binarization will speed up the operation, can it achieve a better trade-off in performance and efficiency than a real-valued efficient SR network. It is suggested to compare with some efficient SR networks [5, 6, 7] in terms of Params, FLOPs, inference time and performance.

> 1. Flexible Residual Binarization for Image Super-Resolution. ICML24.

> 2. Q-DM: An Efficient Low-bit Quantized Diffusion Model. NIPS23.

> 3. Binarized Low-light Raw Video Enhancement. ICCV23.

> 4. Binarized Spectral Compressive Imaging. NIPS23.

> 5. Efficient long-range attention network for image super-resolution. ECCV22.

> 6. DLGSANet: lightweight dynamic local and global self-attention networks for image super-resolution. ICCV23.

> 7. Feature modulation transformer: Cross-refinement of global representation via high-frequency prior for image super-resolution. ICCV23.

**Questions:**

See Weaknesses.

**Limitations:**

Limitations were discussed.

---

> ### Author Rebuttal · Authors · 2024-08-06
>
> # Response to Reviewer mEsa (denoted as R1)
>
>
>
> `Q1-1` Lack of discussion with some related works[1, 2, 3, 4], in particular [1] which is also for binary SR networks. Please analyze and discuss the differences with [1,2].
>
> `A1-1` Thanks for your advice. We add more analyses and discussions of related works and incorporate them into our paper.
>
> **FRB [1]**
>
> - **Differences:**
>   1. ***Application Model:*** FRB targets one-step models (*e.g.*, Transformer); our BI-DiffSR suits diffusion models with multiple sampling steps.
>   2. ***Binarization Approach:*** FRB proposes a new binarization operation (SRB) to reduce performance gaps; our method uses regular binarization operation, while focusing on dimension matching and timestep awareness.
>   3. ***Training Approach:*** FRB uses full-precision distillation (DBT) to guide the binarized network; we train our model using a basic L1 loss for generalizability.
> - **Analysis:** FRB and our approach improve binarized SR models from different aspects. The SRB and DBT proposed in FRB could potentially be integrated with our method.
>
>
>
> **Q-DM [2]**
>
> - **Differences:**
>   1. ***Task:*** Q-DM focuses on low-bit quantization (*e.g.*, 2/4-bit) for generative tasks; we address 1-bit quantization for image SR.
>   1. ***Time Aware Approach:*** Q-DM stabilizes activations over timesteps by training adaptations; we design timestep awareness TaA and TaR to adjust activation distributions.
>   1. ***Training Strategy:*** Q-DM uses distillation to fine-tune binary DM; we train the network with no special training strategy.
>
>
> - **Analysis:** Q-DM focuses on different aspects from our method but could be integrated to explore better quantization diffusion models.
>
>
>
> **BRVE [3]**
>
> - **Differences:**
>   1. ***Fusion Module:*** BRVE uses real-value 1×1 Convs in its fusion block, adding parameters; our CS-Fusion is parameter-efficient.
>   2. ***Temporal Awareness:*** BRVE enhances restoration using temporal redundancy; our approach adjusts activation according to different sampling timesteps.
>
>
>
>
> **BiSRNet [4]**
>
> - **Differences:**
>   1. ***Structure:*** BiSRNet addresses dimension mismatch; our method further considers activation distribution in the skip connections.
>   2. ***Conv Modules:*** BiSR-Conv considers dimensional characteristics in SCI; our TaR&TaA in BI-Conv are timestep-oriented designs.
>
>
>
> `Q1-2` Ablation experiments are not convincing enough. Comparisons with some other activation function or fusion methods [1, 2, 3, 4] should be included.
>
> `A1-2` Thank you for the suggestions. We conduct more ablation studies on activation functions and fusion methods, with settings consistent with Sec. 4.2 (ablation study) of our paper.
>
> **Activation Function:**
>
> We compare our timestep-aware activation function (TaA) with RPReLU and LeakyReLU, which are applied in previous methods.
>
> | Method     | Params (M) | OPs (G) | PSNR (dB) | LPIPS  |
> | :--------- | :--------: | :-----: | :-------: | :----: |
> | LeakyReLU  |    4.32    |  36.67  |   24.92   | 0.0903 |
> | RPReLU     |    4.37    |  36.67  |   29.27   | 0.0337 |
> | TaA (ours) |    4.58    |  36.67  |   32.66   | 0.0200 |
>
> Results demonstrate the superior performance of our TaA.
>
>
>
> **Fusion Method:**
>
> We compare our channel-shuffle fusion (CS-Fusion) with BFB in BRVE [3] and BiFD in BiSRNet [4]. The structures of these fusion modules are detailed in the **attached PDF (Figure 1)**.
>
> | Method    | Params (M) | OPs (G) | PSNR (dB) | LPIPS  |
> | :-------- | :--------: | :-----: | :-------: | :----: |
> | BFB       |    6.38    |  43.11  |   30.53   | 0.0303 |
> | BiFD      |    4.30    |  36.67  |   29.67   | 0.0384 |
> | CS-Fusion |    4.30    |  36.67  |   31.99   | 0.0261 |
>
> 1. Compared to BiFD, our CS-Fusion promotes fusion without increasing the Params.
> 2. Compared to BFB, our CS-Fusion has smaller Params and OPs.
>
>
>
> `Q1-3` It is always known that diffusion models are slow. Although binarization will speed up the operation, can it achieve a better trade-off in performance and efficiency than a real-valued efficient SR network. It is suggested to compare with some efficient SR networks [5, 6, 7] in terms of Params, FLOPs, inference time and performance.
>
> `A1-3` Thank you for your suggestion. We compare our proposed BI-DiffSR with ELAN [5] and DLGSANet [6].
>
> ELAN, lacking official results, is retrained under the same settings as our method, yielding **ELAN***. We test the OPs and latency (simulated by daBNN) with outputs of 3×256×256, and assess LPIPS on Manga109 (×2).
>
> | Method                          | Step | Params (M) | Per-Step OPs (G) | Total OPs (G) | Per-Step Latency (s) | Total Latency (s) | LPIPS  |
> | :------------------------------ | :--: | :--------: | :--------------: | :-----------: | :------------------: | :---------------: | :----: |
> | ELAN* (**Transformer**)         |  1   |    8.25    |      161.24      |    161.24     |        50.61         |       50.61       | 0.0206 |
> | DLGSANet (**Transformer**)      |  1   |    4.73    |      73.52       |     73.52     |        20.07         |       20.07       | 0.0210 |
> | BI-DiffSR (**Diffusion**, ours) |  50  |    4.58    |      36.67       |    1833.44    |        13.00         |      650.09       | 0.0172 |
>
> 1. Compared to real-valued efficient SR methods, our BI-DiffSR obtains superior **perceptual** performance (**LPIPS**).
> 2. Our method achieves comparable **parameters**. Meanwhile, the **per-step** OPs (**36.67 G**) and latency (**13.00 s**) are lower, although the multi-step nature of DM increases total OPs and latency.
> 3. Model compression (like binarization) and sampling acceleration are two **parallel** speedup strategies for DMs. As mentioned in the main paper, our method focuses on the former to accelerate **one inference step**. Thus, we apply the regular sampler (*i.e.*, DDIM, **50 steps**) to show its generality.
> 4. Furthermore, our method has the potential to be used with more advanced samplers to further enhance the efficiency of DMs.

---

> > ### Comment · Reviewer_mEsa · 2024-08-09
> > **Further Discussion**
> >
> > I appreciate the author's efforts for rebuttal and some of my concerns were addressed. But I still have some concerns regarding the comparison of this work with real-valued networks.
> >
> > 1. PSNR as an important performance evaluation metric needs to be included in the comparison.
> >
> > 2. It is neither fair nor valuable to compare the per-step computational efficiency. As a binary network, it is natural for it to be more efficient per step than a real-valued network. However, for this method, single-step diffusion completely fails to achieve the claimed performance. As it stands, this work is far less efficient than an efficient real-valued network. In the future, with the development of specific hardware and frameworks, this work may be able to realize significant efficiency improvements.
> >
> > Overall, after rebuttal, I'm willing to raise my score to borderline accept.

---

> ### Author Response · Authors · 2024-08-10
> **Further Discussion on Comparison with Real-Valued Network**
>
> Dear Reviewer mEsa,
>
> Thank you for your response. We are pleased to have addressed some of your concerns. Regarding further questions about comparisons with real-valued networks:
>
> 1. We re-evaluate the testing results and add the PSNR values in the following table, where our method does not have higher PSNR values. However, it is important to emphasize that diffusion models (DMs) are generative models, which usually do not achieve very high PSNR values. Due to the perception-distortion trade-off, PSNR does not effectively reflect the performance of DMs. Reviewers can refer to the visual results in our main paper and supplementary file, where our method obtains visually pleasing results consistent with the LPIPS comparisons.
>
>    | Method                          | Step | Params (M) | Per-Step OPs (G) | Total OPs (G) | Per-Step Latency (s) | Total Latency (s) | PSNR (dB) | LPIPS  |
>    | :------------------------------ | :--: | :--------: | :--------------: | :-----------: | :------------------: | :---------------: | :-------: | :----: |
>    | ELAN* (**Transformer**)         |  1   |    8.25    |      161.24      |    161.24     |        50.61         |       50.61       |   39.34   | 0.0206 |
>    | DLGSANet (**Transformer**)      |  1   |    4.73    |      73.52       |     73.52     |        20.07         |       20.07       |   39.57   | 0.0210 |
>    | BI-DiffSR (**Diffusion**, ours) |  50  |    4.58    |      36.67       |    1833.44    |        13.00         |      650.09       |   33.99   | 0.0172 |
>
> 2. Providing the single-step efficiency of DMs is to demonstrate the effectiveness of our proposed method, since our method focuses on compressing single-step sampling. As we mentioned, as the framework evolves (*e.g.*, fewer sampling steps), the efficiency of DMs can be further improved. However, this is not the focus of our research. In the future, we will explore applying our method to more advanced architectures to explore more efficient DM.
>
> 3. We totally agree with Reviewer mEsa that with the development of specific quantization-friendly hardware and frameworks, our work may be able to realize significant efficiency improvements. We will explore this direction in the future. Thanks for the suggestions.
>
> Best regards,
>
> Authors

---

### Author Rebuttal · Authors · 2024-08-06

# Response to all reviewers and area chairs



Dear Reviewers and Area Chairs,



We thank all reviewers (**R1-mEsa**, **R2-Z33c**, **R3-KWS7**, **R4-MRyP**) and area chairs for their insightful comments and valuable time.

We are pleased that:

- R2 and R3 appreciate our intuitive motivation and diverse and insightful analyses.
- R2, R3, and R4 acknowledge the novelty of our techniques.
- All reviewers recognize the impressive performance of our method, which surpasses existing approaches.



We have responded individually to each reviewer to address any concerns. Here, we offer a summary:

- We discuss the differences with more **related methods**, including binarized SR and quantized DM.
- We add more experiments, including **ablation studies** and comparisons on the new baseline (**ResShift**) with new metrics.
- We compare our approach with **efficient SR networks** and provide an analysis.
- We add a **preliminary section** to our paper and commit to improving our manuscript.
- We introduce methods to **extend our approach** to new modules, models, and tasks.
- We clarify the **gap** between theory and practice, the **advantages** of the binarization method, the **innovative** aspects of BI-Conv, and the **difference** between TaR and time embedding.
- We address various **detailed questions** raised by reviewers, including actual running speeds, training times, module efficiencies, and ARM hardware types.
- Finally, we include an **attached PDF** as a supplement for some questions.



Thanks again to all the reviewers and area chairs. We appreciate you taking the time to review our responses and hope to **discuss further** whether the issues have been resolved. If you need any clarification, please let us know.



Best Regards,

Authors

---

### Decision · Program_Chairs · 2024-09-25

**Decision:**

Accept (poster)

**Comment:**

The paper received all accept recommendations. The area chairs agree with this recommendation.